# Equine Management in UK Livery Yards during the COVID-19 Pandemic—“As Long As the Horses Are Happy, We Can Work Out the Rest Later”

**DOI:** 10.3390/ani11051416

**Published:** 2021-05-14

**Authors:** Tamzin Furtado, Elizabeth Perkins, Catherine McGowan, Gina Pinchbeck

**Affiliations:** 1Institute of Infection, Veterinary and Ecological Sciences, University of Liverpool, Cheshire CH64 7TE, UK; cmcgowan@liverpool.ac.uk (C.M.); ginap@liverpool.ac.uk (G.P.); 2Institute of Population Health, University of Liverpool, Liverpool L69 3GL, UK; lizp@liverpool.ac.uk

**Keywords:** equine welfare, COVID-19, livery yards, equine management

## Abstract

**Simple Summary:**

The majority of the UK’s leisure horses are kept at livery yards, under the oversight of a livery yard manager or owner. Livery yard managers/owners therefore have a major impact on the potential wellbeing of horses within each yard, but their role has not been previously studied. This study used the COVID-19 pandemic as a lens to view the priorities and decisions of livery yard managers at a time when everything was already subject to change. Yard managers/owners of 24 very diverse yards were interviewed over a period of nine months, repeating interviews quarterly. Discussion forum threads on relevant topics about livery yards during COVID-19 were also collected. These data were analysed using a qualitative approach called Grounded Theory. The results showed that running a yard required a careful balance between conflicting priorities, which made change difficult within each individual yard culture. During the pandemic, maintaining usual horse care routines and standards was prioritised above human health and the business model of the yard, and above horse–human relationships. However, each yard adopted individual approaches to managing human health alongside maintaining equine management. These findings are important for future initiatives which aim to support livery yard management and change.

**Abstract:**

Approximately 60% of the UK’s leisure horses are kept at livery yards under the management and oversight of a livery yard owner or manager (LYO/M), yet their role has received little research attention. This study used the COVID-19 pandemic as a lens through which to view LYO/Ms’ decisions around equine care and management at a time when changes to usual practice were necessary. Qualitative research methods were used. Up to 3 interviews were conducted with 24 different LYO/Ms over nine months (*n* = 48). Discussion threads from open-access UK discussion fora were also analysed. All data were anonymised and analysed using a Grounded Theory methodology. Prior to the pandemic, equine care and management practices varied greatly across yards, and yard cultures were a product of LYO/Ms’ construction of good equine care, their business model, and the need to balance human and equine contentment. The role of the LYO/M was to maintain an equilibrium between those interlinked factors. During the pandemic, LYO/Ms adopted new measures designed to influence the movement of horse owners and other people on yards to minimise the risk of COVID-19 transmission. During this time, LYO/Ms reported prioritising equine wellbeing by limiting change to equine routines and management wherever possible. Instead of altering equine management, there was an expectation that the lives of humans would be moulded and re-shaped to fit with the government COVID-19 guidelines. These results highlight the importance of routines, traditions and cultures in each individual yard. Maintaining the standard of care for the horse was prioritised regardless of who provided that care.

## 1. Introduction

COVID-19 disrupted many aspects of human life during 2020. This disruption yielded an opportunity for reflection about how we structure and prioritise our lives; for example, which aspects of daily life we changed, and which aspects we considered unalterable. In this paper, we consider the impact of the pandemic on the management of horses and ponies in livery yards, and what this tells us about livery yard owners’ and managers’ perceptions of what constitutes good horse care, people management, and management of the livery yard as a business.

The majority of the United Kingdom’s (UK) horses are kept away from the owner’s home [1], predominantly on livery yards [2], where owners pay for some elements of their horse’s care and management This service could constitute simply renting a space (stable and/or field), an entire service package of care for the horse, or anything in between. In each yard, the livery yard owner or manager (subsequently referred to as an “LYO/M”—Livery Yard Owner/Manager) can control many aspects of the horse’s management. Previous research has found that LYO/Ms not only control the more obvious items such as the field space the horse occupies, the horses with which it shares its field, and its stabling, but in some cases may also determine what the horse is fed, which professionals can visit, and what veterinary care it receives [3].

LYO/Ms’ decisions therefore affect the wellbeing of the horses in their yard. Some LYO/Ms may promote some aspects of equine wellbeing, such as helping owners to care for their horses or reach their goals [4], and supporting owners with care of the ageing horse [5]. However, previous research has also highlighted tensions between horse owners and LYO/Ms in equine management priorities. Many horse owners reported difficulties managing equine obesity because of the rules designated by the LYO/M: for example, not being allowed to implement changes in order to effectively manage overweight horses (such as using electric fencing to reduce grazing) [6]. Similarly, research has found that many owners of liveried horses are uncomfortable with the worming practices in place at their yard [7].

There remain serious concerns about equine welfare in the UK; unrecognised stress, obesity, and delayed euthanasia are examples of serious welfare concerns which have been recognised by stakeholders as urgently needing to be addressed [8,9,10,11]. Given the perceived prevalence of these welfare issues, and the high number of horses kept at livery yards of one sort or another (thought to be over 50% [2]), it is clear that the role of the LYO/M is of particular importance to understand; such knowledge is valuable in finding ways to ensure or improve the welfare of UK leisure horses.

Despite the importance of their role in equine welfare, livery yard owners and managers have been overlooked in terms of previous research. The industry is almost entirely unregulated. A few recognition schemes exist; for example, the British Horse Society (BHS) has a livery yard approval scheme [12], but the yards who pay for approval (approximately 950 at the time of writing) are likely to represent only a small fraction of total numbers of yards operating across England. As a result, each yard in the UK represents its own microcosm, with its practices built over time according to the LYO/M’s ideas of how horses should be kept, cultural and traditional influences as well as the availability of land [4]. Due to the lack of regulation, LYO/Ms are free to manage their space as they choose. Further, given that the number and diversity of horse owners outstrips the supply of livery yard premises, many suggest that they are also uninhibited by competition. If horse owners leave because practices at one yard are unsuited (perceived or real) to the optimal welfare of their horse, there will be other horse owners to fill their space. Therefore, if we are to improve common equine welfare issues on any wider scale, it is important to understand how LYO/Ms make their decisions, in order to support them in making changes that promote good equine welfare practices.

The COVID-19 pandemic and its associated lockdowns had as much of an impact on the equestrian leisure industry as on any other. Most horses are kept away from home, requiring horse owners or livery staff to travel from home, or designate their horse’s care to someone else. While travelling for “animal welfare” purposes was allowed under UK guidelines (i.e., to ensure that horses were cared for), there remained aspects of these rules which were unclear—for example, whether exercising the horse remained permissible and whether owners should be allowed to visit their horses and on what basis this should be determined. Many yards made alterations to their practices in order to limit movement on the yard and to allow for social distancing while on the yard [13].

The COVID-19 pandemic therefore represented an unprecedented opportunity to identify how yard owners and managers balance what they perceived as needs of the horses with the rights and responsibilities of the owners, when some of the owners’ rights and responsibilities had been restricted and curtailed through government intervention. This study aimed to study the way in which livery yard managers negotiated the COVID-19 pandemic, and the changes that they considered appropriate to ensure both human and horse health.

## 2. Materials and Methods

In order to explore the issue of equine care during the COVID-19 pandemic, this study used qualitative methods. Qualitative methods are ideal for situations where little is known about the issue in question, and exploration is required [14]. This study incorporated interview data from 25 LYO/Ms interviewed throughout 2020, as well as observational data from open-access UK discussion fora. This study was approved by the University of Liverpool’s Veterinary Ethics Committee (VREC953 21 April 2020).

Livery yards were recruited predominantly through social media, as well as via “snowballing” (existing participants or contacts suggested to their networks that they might participate). It was explained to participants that this study was exploring how livery yard managers were making decisions around equine care and wellbeing during the COVID-19 pandemic, but that the interviews would also cover how they ran their yard before the pandemic. Informed consent was taken prior to beginning each interview. The interviews were semi-structured, following an interview guide available in the Appendix A, though participants were free to lead the direction of discussion. Follow-up interviews were conducted every 3–4 months with yards who wished to take part in them. Follow-up interviews covered any changes made in the intervening months, as well as picking up on any issues not covered in the first interview. Twenty four yards participated in the initial interviews, 15 yards participated in a second follow up call, and 9 in a third follow up. This led to 48 interviews overall, with yards representing a diverse mix including DIY, full and part livery; reschooling; retirement; and livery alongside another business (e.g., riding school, equine assisted learning facility). The yards also incorporated a mix of LYO/M participants who both owned and managed the yard, rented the premises from a yard owner but managed it themselves, or owned the yard and sublet to a yard manager (Table 1).

Interviews were conducted over the phone or via a secured audio-call facility (e.g., secure Zoom rooms). Participants often also shared photographs, documents (such as yard contracts and COVID rules), as well as their websites and Facebook pages, with the interviewer. Interviews were then audio recorded and transcribed, and anonymised: all names, locations, and other identifying data were removed.

In order to collect observational data from horse owners as well as LYO/Ms, discussion fora were also collected between April 2020 and January 2021, from two open-access discussion forums. The forums were selected based on having primarily UK-based discussions, being open access with content publicly available, and being regularly used (new posts at least every day). The forums were searched monthly with the keywords “COVID”, “pandemic” and “livery”. Threads were incorporated into the project if they were focussed around the issue of caring for horses during the pandemic, particularly for those at livery. These threads were usually posted by the equestrian community generally, rather than by LYO/Ms. Eight threads were included in the analysis, predominantly covering how yards had dealt with the pandemic, and how participants felt about the changes. These threads were downloaded and anonymised.

Anonymised interview and discussion forum data were placed in nNVivo 10 to facilitate analysis. Analysis was performed according to a Grounded Theory methodology, using the approach described by Charmaz [15,16]. Analysis was performed while data were concurrently collected, and the initial analysis informed subsequent data collection (for example, specific yard types were sought after initial rounds of recruitment). Data were analysed according to line-by-line, iterative interpretation, and were subject to constant comparison with new data.

In order to code the data, the lead researcher initially read through the documents to familiarise herself with them, and made notes about themes or ideas that emerged. Following this initial reading, closer reading and an initial coding exercise led to the creation of early codes which were “close” to the data—for example, closely described the themes in each narrative. For example, an early theme of “negotiating human-human contact on the yard” might apply to sections where respondents were speaking about this topic. As more data were added to the analysis, these themes were constantly checked, and sometimes the code label (its name) was altered, codes were combined together, or split apart. Parent themes were also added, according to important concepts. These themes and their contents were constantly checked against new data, until the final framework was considered to represent the experiences described in the participants’ narratives and forum data. That framework and its results are described below.

## 3. Results

This paper reports decision making around equine management on livery yards during the COVID-19 pandemic. It predominantly focuses on the initial lockdown, when most changes were made. These data were refined into a framework (Figure 1) which shows how the themes interpretation of risks from COVID-19 and interpretation of guidance combined with the LYO/Ms’ ideas about managing the yard, equines and clients, led to a change in their role from management of the status quo, towards strategizing, communicating and leading during a time of transformation because of the COVID-19 pandemic. At this time, the requirement to satisfy the needs of the business, clients and—most importantly—the horses led to three main management changes—reducing exercise, streamlining care (making horse care quicker, easier, or less labour intensive), and increasing turnout.

Livery yard managers perceive their usual role as being a responsible manager who will maintain three interlinked outputs: a stable viable business; client satisfaction (including health and wellbeing); and equine wellbeing. The threat to human safety from COVID-19 resulted in a period of change on the yard in order to renegotiate management around this new risk. Without industry-specific guidance LYO/Ms needed to balance the interpretation of formal guidelines with their own perception of the risks of COVID-19 and the risks of changing their management on the horses. Each LYO/M found individualised solutions to maintain an equilibrium to suit their business, clients, horses, and their own health. For most yards, this level of change was unprecedented, and studying yards at this time provided considerable insight into the role of the LYO/M more broadly.

### 3.1. The Responsibilities of Management—Being a Livery Yard Owner or Manager in Pre-COVID-19 Times

LYO/M narratives highlighted that their role in pre-COVID-19 pandemic times was to maintain an equilibrium between three sometimes competing goals, as depicted in the image. In order to maintain the business, satisfied customers were needed, and satisfied customers were often thought to arise from happy horses:

Participant (P)6: *If you look after people’s horses and you’re nice to them, and you make the environment really friendly, you’ll have good clients, who stay all the time*. [manager of a 34 horse yard, offering services ranging from DIY to full]

P14: *If the horses are happy, the owners are happy*. [manager of a 63 horse DIY yard]

Nevertheless, the need to maintain client satisfaction for a number of different clients, who all wanted individualised care, was frequently mentioned to be the most difficult part of running a yard. As a result, LYO/Ms described themselves as needing to be confident in their decisions and act “like a dictator”, laying down clear rules and regulations about how clients should act on the yard. If clients did not want to act within those rules, then it was better for them to leave:

P7: *if they don’t like your rules, then go somewhere else where there are no rules*[owner and manager of a 15-horse livery yard catering to DIY and part livery]

P16: *you take no prisoners. If someone upsets you, you kick them off. Then they learn that I don’t mess around, so they don’t mess around*[manager and owner of 2 livery yards, one holding 40 horses and one holding 12, all DIY]

To some extent, the responsibility of “keeping clients safe and happy” could sometimes be stretched or de-prioritised compared to the other two areas of responsibility: unhappy or uncompliant clients could be easily replaced.

Therefore, each yard functioned as a microcosm of what LYO/Ms believed constituted an appropriate business model and good horse-care practices. The ideal was to achieve a match between the clients’ expectations of good horse care with the practices offered by the LYO/M. This was the case whether the respondent owned and managed the yard themselves, owned the yard and sublet to another manager, or managed the yard but rented it from a yard owner. In the latter two situations, owners tended to sublet to people who upheld their views of ideal horse care and yard maintenance, thus furthering the proliferation of like-minded communities of individuals within a yard environment.

As well as navigating the potentially conflicting needs of owners, horses, and the business, LYO/Ms allowed for some level of change (e.g., changing clientele, horses being sold or replaced, seasonal changes) within each yard. However, aside from these changes and the occasional improvement to facilities, it was rare for LYO/Ms to describe major changes which occurred on their yards with horse-care routines and practices reported to have been maintained over many years. The exception to this was two yards who had altered the yard from traditional management to provide what they described as “horse-centric” livery (e.g., track systems or “natural” horse care; please note other yards also used such systems, but had purpose-built premises for this rather than adapting from an existing thriving business which used traditional management methods), who had made changes not only to their business model and horse care, but also described constant “tinkering” with their set-up to adapt according to conditions. However, these yards were in the minority. LYO/Ms described customs and traditions which persisted regardless of other factors (for example, the hours of turnout in summer and winter were fixed in many yards, changing over on certain dates, rather than according to conditions). In pre-COVID-19 pandemic times, LYO/Ms therefore described having created an environment for horse care, and then maintaining the status quo by directing clients to act within a designated set of expectations. However, all LYO/Ms’ roles changed during the COVID-19 pandemic, as change was negotiated.

### 3.2. The Responsibilities of Leadership during the COVID-19 Pandemic

The COVID-19 pandemic pushed human wellbeing and safety to the fore. LYO/Ms described needing to flexibly adapt their yard environment, their own expectations, and their management of clients and horses alike in a very short time period.

Decisions about equine care were considered to be individual to each yard manager, meaning that no “blanket” rules could be adopted; instead, it was the role of the LYO/M to act as a leader and weigh up multiple factors including the yard set-up, the horses, the clients, and above all the LYO/M’s perception of the various risks involved with having, or not having, people visit the yard:

P6: *Every client, every horse and every yard have different things to deal with…It’s all very well saying “all close down.” Yes, that’s easy for you to say, because you’ve got a DIY yard with half a dozen hairy cobs that you could chuck in a field. Again, everybody’s circumstances are so different, you have to interpret it for yourself.*[manager of a 34 horse yard, offering services ranging from DIY to full]

LYO/Ms described that they were aware of the need to find a solution which suited the circumstances of their yard. They described that they also needed to mitigate against a risk which affected them personally: horses on the livery yard are the legal responsibility of the LYO/M [17]. Therefore, they were aware that, if staff or livery clients became sick, LYO/Ms would have sole legal and ethical responsibility for the wellbeing of all the animals on the property. Given that the normal workload was often distributed across several staff members (for full livery) or a great number of livery clients (for assisted/DIY livery), this would present a significantly higher workload. LYO/Ms therefore needed to construct a method for managing the horses on the yard, whilst maintaining safe hygiene measures for staff, clients, and family members (who often lived on the yard premises) alike, in order to protect themselves from having to care for all the equines on the yard:

P4: *I thought, “This could actually get really nasty because potentially here I’ve got 14 horses that I’ve got to look after on my own if the government goes as far as they have in a couple of the other countries.”*[manager of a 12-horse DIY yard (also keeps her own 2 horses at the yard)]

P16: *I was worried it was going to come to owners not being allowed to drive to their horses, but thank God, it didn’t come to that, because I did not relish the thought of looking after 40 horses on my own*[manager of 2 livery yards, one holding 40 horses and one holding 12, all DIY]

Discussion forum respondent: I would not close my yards if the livery’s want to come, as I am a one man bandit and could not muck out and care for all mine and the l part livery and the diy. [Livery owner on discussion thread about how livery yards are managing during the COVID-19 pandemic] (grammar and spelling of original post retained)

Therefore, it was in the LYO/Ms’ interest to maintain client and staff visits to the yard if possible, both to maintain positive relations with their clients, and so that the LYO/M did not have to care for all the horses.

Maintaining client access to the yard was also key to maintaining client satisfaction and wellbeing. In pre-pandemic times, LYO/Ms rarely discussed moral or ethical decisions that they needed to make, aside from occasional concerns about horse wellbeing because of uncompliant liveries on their yard. However, the COVID-19 pandemic brought to the fore a set of complex moral dilemmas around client health and safety in terms of COVID-19 transmission risk, versus human emotional wellbeing, given that clients wanted to visit the yard for their mental and emotional health. Many LYO/Ms described feeling under pressure to also navigate the emotional and mental health of their livery clients, by allowing them to spend time at the yard:

P9: *It’s, sort of, that balance between mental health, I suppose, and physical, isn’t it? She said, “I can’t stay in the house. I have to be able to get out.” I didn’t want to stop her but, equally, I don’t want to put her at risk.*[in relation to a particularly vulnerable, shielding client, who wanted to visit her horse] [manager of 27 horse DIY yard]

P6: *I’ve had clients who have said, “My mental health situation is such that not seeing my horse-” and I’m thinking, “Don’t do that to me, I’m not a professional psychiatrist, it’s not my situation to judge that”*[manager of a 34 horse yard, offering services ranging from DIY to full]

Managing this moral dilemma led to LYO/Ms making adjustments which would allow them to maintain client visits to the yard where possible, whilst creating rules and guidelines which would alter the usual yard practices to mitigate the risks from COVID-19:

P7: *I thought I’ve got to get a set of rules in place for everyone, so that everyone knows. So I started off with this set of rules*.[manager of 15 horse yard offering DIY livery and part livery]

However, unlike pre-pandemic rules, the rules initially imposed by LYO/Ms often changed significantly over short time periods, in response to compromise, co-operation and negotiations with the clients. For example, one yard arranged a yard council with representatives from each type of livery (full, part and DIY) who could work with the LYO/M to find suitable compromises for horse care and client satisfaction.

### 3.3. Interpretation of Risk: Prioritising Human Health and Safety

In pre-pandemic times, it was common for LYO/Ms to follow the same traditions and annual practices for many years. However, the new risk to human health from COVID-19 was particularly unusual; respondents suggested that human health was an area which was prioritised below horse health and safety, both by YLO/Ms and in the general equestrian community:

P14: *If it was Strangles, I think people on the yard would be a lot more scared…they’re more concerned about their horses than they are themselves*.[manager of 63 horse DIY yard]

LYO/Ms’ perception of the severity of risk from COVID-19 was particularly important in how LYO/Ms thought about managing COVID-19: the more concerned the LYO/Ms, the stricter the levels of lockdown:

P4: *I’m an ex-nurse and have been in pharmaceutical research so I understand what’s going on. That’s probably all the combination that has helped to make me very proactive about it, that plus the fact that my partner has had it.*[manager of 12 horse DIY yard]

P15: *Quite truthfully, I think the hysteria around COVID, having had it and all the rest of it, is—social distancing is a sensible thing, lockdown is not. I’ve had it, I’m not going to get it twice. …The decision [to allow people from other yards to pay to use the cross country course] is based on the fact that I’m probably more gung ho than I should be, because I don’t think I’m going to get it twice, certainly not within the interval that we’re talking*.[manager of competition centre which is sublet to three yard managers: holds 80 horses total on full livery]

In the first instance, P4 used her medical knowledge and personal experience to share her concern about the risk of COVID-19 with her liveries, who she said therefore listened to her and were “quite happy” with the changes she put in place, even though she was clear with them that those changes would be long term. In the case of P15, however, the perception of the risk of COVID-19 was much lower, and he was happy to open up his competition centre as soon as official guidance would allow, though he put in place strict distancing measures in order to do so.

Another place where new levels of risk interpretation around human safety were required was in the call to “protect the NHS” (National Health Service) by not taking unnecessary risks which could potentially result in hospital visits. Given that horse riding (or care) is, by nature, a particularly risky activity with a high injury burden [18,19,20], there were calls in the equine community to either stop riding outright, or at least reduce more risky activities (such as jumping). Six LYO/Ms felt that an outright ban on riding was most appropriate, as in P8’s case:

P8: *We felt we had to support the NHS by not letting people ride in case they came off.*
[manager of 12 horse yard offering mainly part livery, with some DIY]

However, the remaining LYO/Ms suggested that the decision of whether people should ride was not straightforward, given the need to weigh the risk of personal injury from riding incidents, against the risks that arose from stopping people riding: upsetting their clients, emotional distress, and having horses fall out of routine:

P14: *There were some opinions that horses shouldn’t be ridden at all but we, sort of, thought, “Well, it’s all part and parcel of the horse’s welfare.” And, particularly, coming in to spring, they still needed exercising and working. You’ve got to balance mental health of the clients and it’s a whole mixture, really*
[manager of 63 horse DIY yard]

As a result, most yards “recommended” not riding, or else reduced access to areas which presented a higher risk such as jumping fields. One LYO/M did not ban jumping, but instead placed new pole exercises in the arena each day to encourage her liveries toward lower-risk flatwork rather than jumping.

### 3.4. Interpretation of the Guidelines

In order to institute new COVID-19 safe guidance on their yard, LYO/Ms had to evaluate both the generic advice from the government and the more specific, but allegedly confusing, guidance from the equestrian authorities such as the advice from British Equestrian (BE) and the British Horse Society (BHS).

P10: *I found that when lockdown was announced, the guidance and support was absolutely shocking for livery yard owners, it was just horrendous….worse than useless. I think they caused more problems than what they helped with….they just left us high and dry with absolutely nowhere to turn, and we had to cope with our liveries, and it was horrendous.*
[manager of 36 horse yard offering DIY, part and full livery]

Guidance from the first lockdown suggested that travel was only permitted for “essential” reasons, which included animal welfare and caring for livestock [21]. This meant that clients who did not care for their horses’ day-to-day needs (e.g., clients with horses on full livery) should not visit the yard. However, as described earlier, LYO/Ms were under pressure to navigate their ongoing business viability and the emotional wellbeing of their clients alongside the risk of COVID-19, and therefore this guidance was interpreted differently by different yards: some simply banned full livery clients from visiting their horses for a period of time, and others allowed clients to visit their horses as part of their “daily exercise”. As P6 suggested, the guidance was open to interpretation, leading to different yards responding differently:

P6: *These are not laws, they’re guidelines, and I think you have to think for yourself and be responsible. We’re not looking for loopholes, we’re interpreting, and people interpret differently.*
[manager of 36 horse yard offering part, full and retirement livery]

However, the guidance presented a particularly difficult situation for LYO/Ms whose yards had both horses kept at full livery (where clients technically had no reason to visit) as well as horses on part or DIY livery, where clients could visit as an “essential” journey to care for livestock.

P6: *We then fall between two stalls, because we are basically full and part livery. So it is not essential travel, it’s-They’re full livery so we can actually look after the horses, there is no reason for anyone to be here at all, is the original guideline. So that put us in quite a difficult situation*
[manager of 36 horse yard offering part, full and retirement livery]

Yards which catered for both full and DIY livery, therefore, sometimes found themselves in a complex predicament where full livery clients were effectively banned from visiting, but DIY liveries “*spending huge amounts of time at the yard, riding and bathing horses*” [discussion thread] at the yard, under the guise of “looking after livestock”. This led to the need for compromise; relaxing the guidance for full liveries, while restricting DIY liveries. This meant needing to either re-interpret or go against the official guidance. One yard temporarily swapped her full livery clients onto a DIY contract, risking her business, so as to enable full liveries to visit the yard as often as DIYs.

### 3.5. How and What to Change? The Difficult Task of Readjusting Workload While Maintaining Human Safety and Equine Wellbeing

Taking into account the complexity of the moral and legislative dilemmas about how to manage a livery yard during the COVID-19 pandemic, yard managers were left with a range of practical considerations to balance with the priority of maintaining equine wellbeing. These included how to minimise the human workload (so as to reduce footfall on the yard and therefore risk of disease transmission, and also to reduce the burden if the LYO/M was left with sole responsibility for horses in their care), how to keep people safe from contracting the disease while on the yard, as well as improving the general safety of both the LYO/M and clients so as not to burden the NHS.

Participants’ narratives made clear that maintaining current equine management and welfare was one of their most significant priorities, and that human lives could be readjusted to account for the changed practices needed to achieve standard equine care, while maintaining “COVID-safe” practices such as social distancing. Therefore, the disruption to the LYO/M, staff, and horse owners themselves were often huge, while the disruption to the horses’ lives were less apparent. There was no question in LYO/Ms’ minds that this was how it should be; when asked why they chose to work in this way, they suggested:

P14: *it’s the welfare of the horse that’s our priority. We didn’t want the horses to, sort of, suffer or change their routine in any way. It was up to the people to sort themselves out, really.*
[manager of 63 horse DIY yard]

P4: *As much as possible I try and run this place for the horses. Me first and then the horses because I have to be able to stay sane. For me, the important thing is, as much as possible, that the horses have as much routine as possible. Within that then okay, how can we do things safely whilst maintaining things as much as possible for the horses.*[manager of 12 horse DIY livery]

Therefore, LYO/Ms predominantly sought ways that they could maintain their usual routines and standards of equine care, but with increased human health and safety measures and a reduced workload.

#### 3.5.1. Streamlining Care and Reducing the Workload

LYO/Ms described that finding ways to reduce the workload at the yard would have multiple benefits: it would ensure minimum client time was required at the yard, as well making it easier for them as manager, if they ended up needing to look after all the horses for any reason. However, given that LYO/Ms felt it was important to maintain equine welfare standards and, as far as possible, normal care practices, LYO/Ms described three main options:Changing the carer of the horses (for example, one client could turn out two horses rather than having two separate owners come to the yard), OR the LYO/M/staff could look after the horse at one end of the day.Considering increased turnout time in fields.Reducing exercise time of horses.

Altering the carer of the horse was the least disruptive option, and as such was the first port of call for many yards. LYO/Ms described lowering their usual costs for assisting owners with horse care, so that owners would be more likely to take this option. Three DIY yards encouraged “buddy systems”, so that each owner would pair with another, so that each individual would only go to the yard once a day, but would care for two horses when they did so:

P10: *I encouraged them to buddy, so, that’s one half of the buddy would come up at the beginning of the day, and one would come up at the end of the day. They could only buddy with somebody on their block, because nobody was allowed in each other’s block*[manager of 36 horse yard offering DIY, part and full livery]

This strategy had the added bonus that, if one of the “buddies” were to become ill, their partner would already be familiar with the day-to-day care of that horse.

One livery yard went further in streamlining care, suggesting her liveries should make sure that their horses were used to having as little intervention as possible, in case of the pandemic worsening:

P16: *I was saying, “The worst-case scenario is, you’re not going to be allowed to come up.” They were still all in rugs, because it was wet, wasn’t it, and cold? But they were out most of the time, 24/7, thankfully. But I was like, “I want them with rugs off, so please don’t go reclipping them. No other weights for their rugs now. Get them toughened up a little bit, so that if push comes to shove and you lot can’t come up, I can just do drive-by checks of naked horses, check their troughs, throw their hay in from the back of the van and do it.”… they didn’t do anything. I couldn’t see people taking their rugs off or moving their fences. Nobody changed anything*[laughs] [manager of 2 livery yards, one holding 40 horses and one holding 12, all DIY]

Streamlined care meant that the horse’s care needs were ensured, but consistent presence of the owner was not considered by LYO/Ms as central to the horse’s day-to-day wellbeing. However, disparity arose as owners put pressure on LYO/Ms to visit the yard for their own personal emotional wellbeing, clarifying that owners and LYO/Ms consider that perhaps the horse–human relationship is more important to the human, than it is for the horse.

P2: *I mean, nothing’s really changed as far as the welfare of the horses with COVID, I mean I don’t really have any horses that seem to be struggling with not seeing their owners.*
[manager of 10 horse yard offering assisted DIY, part and full livery]

In maintaining welfare during a pandemic, facilitating horse-owner interactions was viewed as optional, while maintaining usual equine care and routines was considered important.

#### 3.5.2. Increasing Turnout Time

Given that 16 of the 24 livery yards used stabling for at least part of the day all year round (usually, stabling during the day in summer and during the night in winter), increasing turnout time was one potential way of minimising human workload whilst maintaining equine welfare standards. In this context, horses were perceived as having everything they needed when turned out. Turnout was seen as part of good welfare, and a time when horses could be allowed to “be horses”, as well as enjoy rest and social time:

P1: *there’s lots to forage for out there, there’s natural shelter, and shade. We thought if something happens to us, we know our horses can pretty much survive for a couple of days, it was our contingency, and also it lightens the workload on us.*
[manager of equine therapy centre which has five livery horses kept at full livery on a track alongside therapy horses]

Therefore, the eight yards which already had turnout at all times, or provided free-choice turnout and shelter access, did not need to make any changes to their equine management:

P17: *no drastic changes, because they’re all out anyway it wasn’t like we had to turn anything out —it was all the same*. [manager of 14 horse track system, offering full livery]

While human intervention was required for turned-out horses (for example, to check health, give supplementary feed, check water, and remove droppings), none of those tasks had the urgency of the tasks needed for a stabled horse, enabling strict routines to be relaxed.

However, the decision of whether to increase turnout time for part-stabled horses was not straightforward, and thus this issue highlights the existing dilemmas which frequently occur on yards about the amount of time spent in fields versus stables.

Firstly, all yards had a balance between the number of horses and the way in which they used their paddocks: stabling part-time not only protected the horses, but was also one means to protect the land. Therefore, yards which chose to increase turnout time were risking their usual paddock maintenance. Eleven of the yards which commonly used part-time stabling chose not to take this risk of increasing turnout time, while others felt it was permissible for a short time:

P4: *It’s going to hammer my summer fields and they’re not going to get much of a chance to recover because of course during the winter grass doesn’t grow. I’m not quite sure what will happen with that later but it doesn’t matter. For the time being, the important thing is that we keep everybody safe. How we figure that out later, we’ll just have to figure it out*. [manager of 12 horse DIY livery]

Secondly, while horses were perceived as having everything they needed while turned out, in other ways turnout was also seen as potentially posing a risk to equine welfare: horses were considered to have the potential to injure themselves, become overweight and laminitic, or have other issues as a result of additional time turned out:

P14: *When it kicked off, some people were saying, “Well, can we just send the horses out in the field until this is over?” And I wasn’t prepared to let them do that, for reasons that, at that time, the fields were very wet. The horses aren’t used to being out. I do know of one place that did do that and I know that one of their horses had a horrific accident and would have died if it hadn’t been for a passer-by.*
[manager of 63 horse DIY yard]

P6: *Some horses, yes, chuck them in a field for a few weeks, they’ll be fine. But others, if they’re not ridden, they’re going to be a bloody nightmare to handle on the ground.*
[manager of 34 horse yard offering DIY, part, and full livery]

As a result, simply turning all horses out full-time was not considered to be a good option for the yards which commonly used part-time stabling. This was often considered on a case-by-case basis, depending on the horse’s health and behaviour, and the severity of need:

P9: *I didn’t want to scare people but, basically, I said, “We’ll have to look at how many people we lose. If we hit a number of people that…” Say we hit 50% of the people that would normally come to the yard were all in isolation, we all agreed that we would muck in together. Then I said, “We would have to then prioritise, so…” Probably 50% of the horses on the yard, including mine, could be chucked out and would probably come to no harm. We would do that, then we would have to focus on the laminitis horses. It’s just the time of year, isn’t it? It couldn’t be a worse time of year with the spring grass coming through*. [manager of 27 horse DIY livery]

These discussions highlight the extent to which human intervention and management was considered to be necessary in maintaining equine wellbeing, even during a pandemic.

#### 3.5.3. Reducing Exercise

Reducing exercise not only lessened the risk of riding-related activities, but also the time people spent at the yard, and their contact with people, structures and objects: tack rooms were perceived as potentially risky areas. Horses not being ridden or “roughed off” were considered to need less attention in terms of rugging, grooming and maintenance. However, as with additional turnout, reducing exercise was seen as potentially problematic for equine wellbeing, and hence was not considered to be an appropriate option for every horse. Exercise was constructed as keeping horses occupied and stopping them from becoming bored and injuring themselves, and also a means to stopping horses becoming overweight and developing laminitis:

P3: *realistically, some of the horses have to be worked, especially the thoroughbreds because they just get out hand. (Laughter) They’re not playing polo at the moment, even though they’re in gentle work, they’re still hooning around the field. They get bored. I mean, they get seriously bored, but whereas the little cobby pony that isn’t fit, can I say it’s essential that she’s in work because we can manage her weight differently? It’s just making that call.*[manager of 6 horse reschooling and full livery yard]

P17: *we have two horses including my horse who are quite overweight, well not now because he’s on the track, but he’s prone to it so I want to keep him in work. And another mare who was the same, so that was quite difficult to sort of get your head round because it’s kind of essential that they don’t get overweight again … I then just worked the mare for her instead, to keep her ticking over so she didn’t just get grossly obese and out of work.*[manager of 14 horse track system, offering full livery]

Exercise was therefore presented as a positive for the horse’s welfare, and this is likely to have contributed to the low number of LYO/Ms who banned riding outright. As the pandemic progressed and lockdown was eased during the summer, some yards suggested people were actually riding more than usual, and that this was a positive for their horses:

P7: *if the horses aren’t getting ridden, they’re going to be on one. I mean, my liveries are riding more than they have ever done. They’ve got all the time, and the horses are happier because they’re being used because they’re being used and they’re going out for rides and hacks… And I think the owners are happier, because they’re actually riding, they’re using their horses, which has got to be better for everybody. And you can take exercise in this, and this is their exercise. So yes, I think everyone is winning, it’s a win-win situation*[manager of 15 horse DIY and part livery]

These results have shown that LYO/Ms found ways to protect their clients while on yards, but within the already-delicate balance of their business, horse and client wellbeing. In doing so, they prioritised changes which had minimal impact on equine wellbeing; highlighting their perception that care but not necessarily horse–human relationships were important to horses, and demonstrating how the careful balance of practices performed on a yard lead to difficulties in making changes to equine management practices such as turnout.

## 4. Discussion

This study demonstrated how the decisions made on livery yards during a pandemic can highlight the priorities and perceptions of LYO/Ms more generally. While livery yards represent a very diverse set of establishments, they are often suggested to be places where management practices can have negative impacts on equine or human wellbeing [22,23]. Therefore, improved understanding of how yards function will help to find ways of sharing best practices and strategies for improving equine or human wellbeing across yards. This study provides an evidence base for further research in this area.

The COVID-19 pandemic provided an ideal opportunity to explore decision making on yards. Given that major changes to equine management are reportedly quite rare on livery yards, the pandemic provided the occasion for traditions and schemas about management to be disrupted. The fact that yards maintained the status quo horse care as far as possible shows how deeply embedded those care practices are in the fabric of the yard culture. Each yard provided the care and management appropriate for the horse (within the limits of its land, facilities, and business model), and these care and management regimes appear almost impervious to radical change. However, when small changes were taken up (for example, yards allowing the use of an extra field for hacking or for additional turnout), those changes were often considered to be favourable changes by LYO/Ms, who suggested that they may continue with the slightly altered practice. Given the issues owners sometimes face in equine management on livery yards described in the introduction section [6,7,22], supporting and encouraging changes which promote good equine welfare practices is important. The findings of this study in relation to avoidance of major change highlight the importance of understanding each yard on an individual level, and in empowering and working with yard owners to make gradual tweaks to practice.

However, one change that did happen on each yard was the move yard managers themselves made from their everyday role of managing the status quo, towards a role of leadership. Their descriptions of their roles before and during the COVID-19 pandemic closely matched the roles described in business studies of management versus leadership: management is considered to comprise “*maintaining efficiently and effectively current organisational arrangements….the overall function is toward maintenance rather than change*” [24]. Comparatively, “*leaders are people who shape the goals, motivations, and actions of others. Frequently they initiate change to reach existing and new goals. Leadership takes much ingenuity, energy and skill*” [24]. Leadership is considered to be a “higher” order skill than that of management. Leadership comprises foresight, planning and management through periods change [25], and using complex personal skills of negotiation and motivation to drive change in a community [26]. During the initial lockdown, LYO/Ms described using people skills more usually thought of as leadership competencies; when clients were unhappy, LYO/Ms frequently described instances of compromise, negotiation and co-operation. The long-term effects of the newly practiced leadership skills will be the subject of further study in the current research project, which will seek to determine whether such skills or changes made on yards will continue over time as the pandemic becomes part of our “new normal”.

One interesting finding which has not previously been discussed in the literature around equine welfare and horse–human relationships, is the divergence between people needing to see their horses, but perceiving that horses do not need to see their people. This study found that it was not considered to compromise equine welfare if the horse’s owner did not handle the horse, as long as its care needs were met by someone. It was, however, considered to seriously compromise human needs if people could not see their horses. Given the recent advancements in study of the horse–human relationship which have found that horses are not necessarily attached to any one caregiver in the way that humans might hope [27], this finding provides an important insight into LYO/M and owner’s ideas of the importance of care from any human versus the company of their owner.

Nevertheless, the horse–human relationship was considered in this study to be very important to the mental health of the owner. Studies of canine–human relationships during the COVID-19 pandemic also highlighted how the benefits of the human–animal relationship and its daily routines to owner mental health were more apparent during the pandemic than ever before [28,29]. Unlike horse–human relationships, in the pandemic most people reported spending more time than usual with their dogs, and therefore feeling closer to them, as well as the relationship helping to mitigate pandemic-associated stress [28]. The impact of horse owning on mental health has received far less study than dog owning, but the results of this study suggest that horse–human relationships may be similarly important to the human side of the pairing.

The issue of human welfare and safety being sometimes de-prioritised compared to equine wellbeing aligns with previous studies, which have shown that danger, hardship and endurance are expected parts of equestrianism [19,30]. As “serious leisure” enthusiasts [4], the equestrian community show commitment to their horses in the face of adversity. The idea of equine care (and not riding, per se) being part of serious leisure is furthered advanced in this study, which found that livery clients were willing to risk their own health in order to visit their horses, even when they were vulnerable and disallowed by their LYO/M. Clients and LYO/Ms all found it hard to prioritise human health, and did so only when they perceived that the animals’ needs had already been met, or under strict rules created by the LYO/M. This finding is similar to the literature around canine–human relationships during the pandemic, which showed that people were willing to put their animals’ health above their own. For example, dog owners said that they would delay hospital treatment in order to care for their pet [31], prioritising the animal over human health.

This study methodology has some limitations. The LYO/Ms who volunteered to take part in this study may not be representative of the overall UK population; as with any voluntary study, response bias is likely to have affected the data. Indeed, the yards who gave up their time described a high degree of professionalism in how they ran their yards. Other studies reported more interventionist practices such as banning riding, or banning owners from seeing their horses [13]. Another possible limitation in the data yielded by this study is the lack of data from male yard owners or managers, since we interviewed only two men. While this is representative of the male:female divide in equestrian sports, future projects which compare attitudes to risk between males and females in relation to yard management would be of particular interest

## 5. Conclusions

Livery yards are important institutions in terms of their effects on equine welfare. This study furthers our understanding of livery yard culture and management, showing how each yard has deeply embedded equine care practices, the importance of which is so entrenched that they changed little, even in the face of an international pandemic. Each LYO/M described making decisions which best suited individual circumstances in terms of the LYO/M themselves, the horses (as highest priority), the clients, and the business; for each LYO/M, it seemed that their way was the only way that they could have managed.

Given that many initiatives and researchers have suggested that improving standards across yards would be one way of improving equine welfare on a population level, this study suggests that supporting LYO/Ms to share best practices and to make small, step-by-step changes will allow them to adapt new initiatives to their own individual situation and balance of business, clients and equine needs.

Moreover, this study has highlighted the complexity of yard management and ownership, and the lack of material, practical or emotional support for those running yards, particularly during times of change such as during the pandemic. In order to support yard owners and managers in changing to improve equine welfare, it is important also to provide support to the other areas of yard management, including the business model, client management, and emotional support.

## Figures and Tables

**Figure 1 animals-11-01416-f001:**
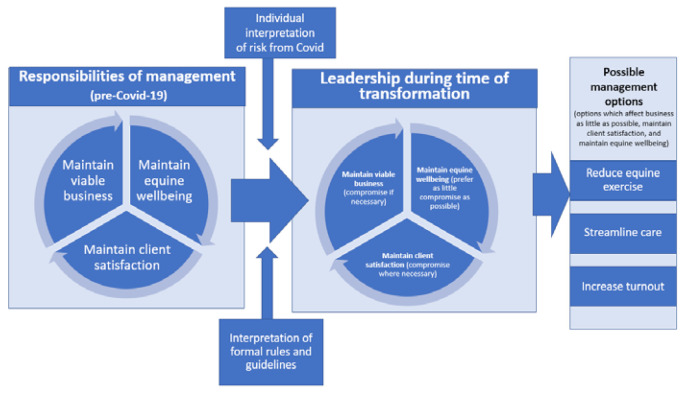
Conceptual diagram of livery yard manager/owner’s move from pre-COVID-19 pandemic responsibilities towards decision making and change due to the pressures of the pandemic.

**Table 1 animals-11-01416-t001:** Participants and livery types involved in the individual interviews, detailing changes made on each establishment.

Yard ID	No of Livery Horses	Type of Livery Offered	Owner or Manager?	Equine Management Offered	Changes for Horses due to COVID-19	Changes for Clients or Business	Effect on Business
1	5	Full livery	Owner, with yard manager employed	Track livery, 24/7 turnout with access to shelter, livery horses kept alongside charity horses used for equine therapy	More space for horses opened up so less care needed–can be left in fields if needed	Owners not allowed onto yard during initial lockdown—one did come anyway. Social distancing and handwash for anyone on the premises	Staff (who primarily work on the charity rather than livery) were furloughed. Worried long term on business because of vulnerable clients
2	10	Assisted DIY/part/full	Owns and manages the yard	Stabled in daytime in summer and night-time in winter	No changes to horse management	1 h/horse/day rule Social distancing and handwashing, etc. No hacking on roads and limited jumping	Some owners put horses on full livery at a reduced rate. Field faeces removal fee waived
3	3–6	Reschooling/full	Owns and manages the yard	Predominantly grass, barn stabling if needed	Limited exercise—no hacking. “Bare minimum” care (turned out)	Liveries not allowed at yard at all (but they come rarely anyway)	Initial reduction in business during first lockdown, but recovered later
4	12	DIY	Owns and manages the yard	Stabled at night all year round	5 horses turned out 24/7 to reduce footfall	Social distancing and handwash, etc.	Emergency care plans put in place
5	20	DIY, part or full, also rehab and schooling livery	Rents the yard and manages it	Stabled in daytime in summer and night-time in winter	No changes for horses	No jumping Wear gloves on yard, disinfectant, etc. Timeslots for visiting	Slower to fill vacancies than usual. Some free care offered
6	34	Full, part and retirement	Owns the yard, oversees its running but also employs a yard manager for day-to-day running	Stabled in daytime in summer and night-time in winter	No changes to management	Initially 1.5 h timeslots but staff were unhappy with more people than usual on the yard—full lockdown for two weeks then timeslots Social distancing, hand washing. Antibacterial gel on doors, etc.	No concerns
7	15	DIY and part	Owns and manages the yard	Stabled in daytime in summer and night-time in winter	No changes to management. More riding than usual—“horses are happier because they’re being used”	Social distancing, hand washing. Antibacterial fel on doors, etc. No visitors	None—people appreciated the changes she put in place
8	12	Some DIY, mainly part livery	Owns and manages the yard	Stabled in daytime in summer and night-time in winter	Turn out 24/7 if possible	Hour time slots. No riding. Social distancing, hand washing. Antibacterial gel on doors, etc.	None
9	27	DIY	Owns and manages the yard	Stabled in night-time in winter; out 24/7 in summer Three are turned out full-time on separate arrangement	Opened summer fields early to allow 24/7 turnout if wanted	Social distancing, hand sanitiser, etc. reduced number people on yard. No hacking on roads—let them ride in winter fields instead. Buddy system	Emergency plans in place to turnout all horses except laminitics
10	36	DIY, part, and full	Owns and manages the yard	Stabled in daytime in summer and night-time in winter	No changes other than no riding, sometimes cared for by “buddy” rather than their owner Considered turning them away but liveries did not like the idea—given that they had another viable option so took that	Social distancing, bare minimum visiting. Stopped riding, stopped kids visiting. “Buddy” system within each separate area. Yard representatives to help make changes	Staff member furloughed
11	60	DIY, part, full and rehab	Rents the yard and manages it	Some are out 24/7 all year; others stabled according to owner preferences	No changes to management—turned out anyway for most part	Specific times for vulnerable people—gates left open, etc. Otherwise, specific visiting times (not rota) Hand sanitiser, etc. Advised not to hack or jump	Kept some horses on full livery at own expense so owners did not come down when ill
12	7	DIY	Owns the yard and manages it	Stabled in daytime in summer and night-time in winter	No changes to management	Small yard (4 owners) so no rota needed. Social distancing, hand gel, etc. Allowed riding, but closed off private hacking	No effect
13	63	DIY only	Rents the yard and manages it	Stabled in daytime in summer and night-time in winter	No changes to management	Social distancing only, no time slots etc	No effect
14	80 (split into 3 seperate yards)	Full only	Owns the yards and sublets to three separate yard managers	Stabled and turned out according to each individual yard manager on the 3 yards	No changes to management	Was up to individual livery managers of the three yards	No effect for liveries, only competition centre
15	40 horses on 1 yard, 12 on the other	DIY only	Owns the yards, manages one and someone else manages the other	24/7 turnout all year on one yard (but with stables which can be used according to owner preference)—other yard stabled in winter	No changes—owners were asked to prepare horses for worst case scenario (e.g., get them used to being together, not wearing rugs and needing minimal care)	Social distancing requested, no hacking in village	No effect
16	14	Full livery only	Rents the land and manages the yard	Track system	No changes except less exercise and horses not seeing their owners during initial lockdown	Complete lockdown—not allowed to visit horses for several weeks; in subsequent lockdowns, owners were allowed to visit while socially distancing	Vacancies harder to fill than usual—otherwise no effect
17	17	Full (training, retirement, etc.)	Owns and manages the yard	24/7 turnout with access to barns, on track and Equicentral (a variation on permaculture) system	No changes	Full lockdown—owners not allowed to visit in initial lockdown but were allowed in later lockdowns if necessary	None
18	58	DIY	Owns and co-manages the yard with his wife	Stabled in daytime in summer and night-time in winter	No changes except less exercise	Social distancing requested, no hacking in village	Actually better off due to break in business rates and grant
19	21	Full and part	Extended family owns the land, and the participant manages the yard	Most horses are on a track system, others have small sections of farm with stables and 24/7 turnout if wanted	No changes except less exercise	Social distancing requested, closed fields for riding. One hour time slots	None
20	22 on one yard, rent another yard of 8 to a professional rider	Mainly DIY but offer services	Owns the land and manages one yard; sublets another yard to another yard manager/professional rider	Stabled in daytime in summer and night-time in winter	No changes	2 h time slots, hand washing and distancing	Took opportunity to re-do indoor school. Lost funds that they usually get from renting arena
21	2	DIY but alongside her own herd	Owns and manages the yard	24/7 turnout and free access to barn	Took one horse on full livery as owner shielding	Social distancing	None
22	40	DIY, half the yard is sublet;does only full livery	Owns and manages the yard	Stabled in daytime in summer and night-time in winter	None	Social distancing	The yard was given a grant, business improved
23	20	Retirement and youngstock—can be DIY or full	Owns and manages the yard	24/7 turnout all year	None	Reduced visits, social distancing	None
24	7	Full and assisted	Owns and manages the yard	Stabled in daytime in summer and night-time in winter	No changes except more exercise	Social distancing, time slots	Riding school activities werelimited but livery propped the business up financially

Abbreviations: DIY “Do It Yourself” livery, whereby the horse owner cares for the horse and the LYO/M provides the space to do so. ”Buddy” system describes owners joining up in pairs to assist one another.

## Data Availability

Data (e.g., anonymised transcripts) will be shared with researchers upon request and approval from ethics committee.

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
