# Peer review of "Equine Management in UK Livery Yards during the COVID-19 Pandemic—“As Long As the Horses Are Happy, We Can Work Out the Rest Later”"

_animals, 2021, doi:10.3390/ani11051416_

Round 1

Reviewer 1 Report

A timely study, taking the advantage of diving in the ongoing situation. Please see bellow some recommendations and comments, with the hope to further help increasing both the value of the paper and clarity for the reader.

Each abbreviation should be spelled out at first usage, even for the consecrated ones such as ‘UK’

Simple summary

Even if the Simple summary is for the general reader, multiple repetition of ‘we’ draws attention and places emphasis on the authors. Please consider using impersonal mode

L12: please consider rewording “everything was already ‘up in the air’”

L15: please insert ‘The’ before “Results’ (specific results of the study)

Abstract

L25: ‘necessary’ may be smoother to read

L27: please consider to change (total 48 interviews) to (48 interviews in total) or (n=48) or (a total of 48 interviews)

L30: the word ‘ideas’ sounds weak; does it refer to knowledge and experience?

L31: please consider to change ‘on’ to ‘in’ after ‘contentment’

L36: ‘instead’ is not clear enough. Please rephrase (LYO/Ms adopted new, movement influencing measures, they limited the horse-management changes to minimal – these are already two slightly/partly contradicting statements. Then the LYO/Ms expectation is stated, for human compliance with the official/legal guidelines, instead of what?)

L36: ‘the’ before ‘humans’ is not necessary (as long as it refers generally to people, humans)

L37: please correct ‘hese’ to ‘These’

L38: please consider to change ‘individual yard’ to ‘each individual yard’ or ‘individual yards’

L38: please consider to insert ‘good’ between ‘a’ and ‘standard’ (to be clearer).

Introduction

L43: please consider to change ‘every aspect’ to ‘many aspects’ or ‘most of the aspects’ (from scientifical point of view the ‘every aspect’ statement is neither proven, nor the aim of the article to prove)

L44: please consider to change ‘prioritise our lives’ to ‘what we do prioritise in our lives’

L45-46: please consider to change ‘could not be altered’ to ‘unalterable’ or a synonym (alternatively ‘which were the aspects of our daily life we changed and which were those we considered unalterable’ or simpler: ‘our readiness or resistance to change specific aspects of our daily lives’)

L46: please change ‘consider’ to ‘considered’. Please use simple past tense wherever the performed research (including results) is presented

L47: please change ‘on’ to ‘in’ or ‘at’

L50-51: citation needed for ‘predominantly at livery yards’

L50: same as for L47

L50: please, here and throughout the paper, follow the requirements of Instructions for Authors regarding the formatting of the reference numbers given in the text and all other requirements stated, as well

L52: ‘alongside other horse owners’ is redundant and even misleading, please remove

L54: same as for L47 here and throughout the paper

L61: please remove the semicolon

L64-69: wording can be changed to be shorter and more efficient, especially in L64-67 sentence

L70-72: with respect, in my opinion, giving four references (7-10) to a reasonably general sentence, without citing the respective titles anywhere else in the paper is not justified and may be perceived as incorrect with regards to the reader. Please correct this here and where it applies throughout the paper

L72: please be more specific regarding prevalence

L73: it would be of a great benefit for the international reader to have a clearer image of the number/percentage of leisure horses kept at livery yards in the UK, please include this information

L79: please spell out abbreviations at their first usage

L80: please be more specific regarding the fraction of approved horse livery yards in England and the overall number

L81: please be more specific if ‘each yard’ refers to the approved or non-approved ones

L93: please insert ‘on’ between ‘as’ and ‘any’

 L98: please be consistent in tense usage within the sentence (parallelism)

L99: please consider to change ‘to practice’ to ‘of their practices’; consider to change ‘footfall’ to ‘circulation’ or ‘movement’ with or without specification of humans; on – in – at

Materials and Methods

L109: please replace ‘COVID-19’ with ‘the COVID-19 pandemic’; here and throughout the paper: please be consistent in the usage of chosen terms

L117: please consider to change ‘that they might be interested in being involved’ to ‘to participate as well’ or similar

L119: same as L109

L124: please correct the beginning of sentence with a number

L132: please consider to use ‘secured’ instead of ‘secure’

L137: please consider to change the word ‘naturalistic’ to a more specific one

L157: please specify what ‘strongly’ represents

Table 1, please be consistent with the usage of periods (.) and also phrasing. The general aspect of the table is untidy. Inconsistencies of abbreviations which, additionally, are not spelled out in the table’s subscript. The content is for the reader, in my opinion it does not have to respect exactly the formulation (wording, phrasing, citation of results) of the working documents. In certain places rewording can shorten, sharpen, and elevate the meaning/content, please consider to do so. Some (but not all) recommendations:

First row: please rephrase ‘if need be.’

Second row: please reword ‘poo’

Third row: contractions should be avoided in academic writing; please rephrase ‘only come rarely anyway’

Sixth row: please correct ‘antibac’

Seventh row: see above

Eighth row: see above

Ninth row: please reword ‘limited people’

Tenth row: ‘No changes other than no riding, sometimes cared for by buddy rather than their owner’ is unclear, please clarify; the same applies for ‘Considered turning them away but liveries didn’t like the idea – had another viable option so took that’; ‘(do your horse and go)’ is redundant; please reword ‘reps’

13th row: ‘it’s been straight forward’ non-academic and redundant

14th row: please reword ‘sep’; reword ‘comp’

17th row: please reword ‘nec.’

20th row: ‘whole’ is redundant; please reword ‘pro’ rider (professional – not necessarily relevant)

22nd row: previous number of horses not relevant; ‘to another lady who does’ irrelevant;

23rd row: please correct capital letter within sentence

24th row: unclear formulation in third column

Results

Understanding fully that the article describes a qualitative research and no statistical analysis was performed/needed, I do consider that numerical results instead of terms such as often, some, most, many, not always, would increase both the value of the paper and the appreciation of the readers, also because those data are available to the authors. Please do consider this opinion for the comments which relate bellow.

L177: please clarify ‘change in their role from management towards leadership’ meaning from the paper’s point of view: move the definition from the Discussion section to the Introduction or Materials and method so as the reader can have form the beginning clearly defined terms

Figure 1 caption: capitalize first letter, don’t capitalize P in ‘Pre-COVID-19’

L184-186: the sentence repeats information given in the figure, please correct

L192: please be more exact on ‘most yards’

L196-199: repetition of same information

L202-204 and in the followings: please consider to use quotation marks for citations. Please be consistent (and grammatically correct) with the use of periods. Personally, I do not see the reason for the use of square brackets instead of simple brackets. At first usage of ‘P’ it has to be spelled out

L210: please consider to change ‘could’ to ‘should’ or ‘must’, ‘have to’

L212: no reason for using italics, please correct

L230: it seems debatable if they ‘had to allow for’ of ‘were subjected to’; maybe consider to use another term

L238: please provide numerical data as scientific result, instead of terms such as ‘minority’

L241: please do not capitalize ‘pre’

L251-252: several repetitions of this statement, please correct

L253-254: redundant sentence (already stated before). Please consider to rephrase the first two paragraphs as the next one (L255-L261) revolves around the same meaning

L268-273: this is not a result, please move to the Materials and Method section

L280: please consider previous comment regarding the use of square brackets and presentation of the participant’s declarations, however, if a parenthesis was ‘opened’ it has to be ‘closed’ too

L286: not clear for the reader what ‘[sic]’ means here and in the followings as well

L289: please be consistent in the usage of COVID-19 related terms

L304-305: please correct punctuation

L308: here and elsewhere, please use ‘to’ to complete the usage of ‘from’ (e.g. from DIY to full)

L317-318: please be more specific and use numerical results instead of terms such as ‘often’ and ‘short time periods’

L322-324: repetition of previously stated concept, please correct

L325-327: consider, instead of indicating here what you are going to describe, to change the title of the following subchapter, to be suggestive enough by itself

L328-330: please correct repetition (information stated previously)

L360: please insert a space after the brackets

L362: please correct repetition of ‘burden’ and regarding the three references given consider the comment from L70-72

L364: please consider comment for L317-318 (‘some’)

L368: see above (‘most’)

L380-382: redundant, please remove (summarizing results belongs to the Discussions section, indicating what will follow is not needed)

L417: please be consistent in font usage (italics)

L433: how many participants’?

L487: please move the definition of streamlined care if not in the Materials and Method section (best), at least before its first mentioning

L520: please use past tense

L545: please use numerical result instead of ‘most’ and ‘some’

L556: please use past tense

L565: please give numerical result instead of ‘not always’

L588: please correct the number of periods

L589: please correct the number of periods, please be consistent in usage of ellipsis (Is it an ellipsis and a period? Not consistent with previous usages of ellipses)

L593-599: paragraph not belonging to Results section, please correct/move

Discussion

L611: please be consistent in the usage of citation marks (at four previous usages ‘status quo’ was not between quotation marks), consider widespread advice to write Latin terms in italics / check Instructions for Authors

L613: please use past tense; please insert ‘According the participants’ answers’ or similar, before ‘each yard’

L619: instead of giving three references for a general and short statement, please explore more the discussion possibilities regarding the respective scientific studies

L624: please correct the font in congruency (italics)

L627-632: please rephrase the citation to avoid the use of italics, ellipses and incongruent periods. Move the definition (see comment for L177)

L644: please correct italics, here and throughout the paper as it is not a conventional way of emphasizing content (assuming that this was the motive for using it)

L655: brackets are not needed (but their content yes)

L661: citation of other studies needed

L676: the small number of LYO/Ms included in the study, compared to their overall numbers in the UK is another limitation

L680: respect the Instructions for Authors for formatting citations

Conclusions

Based on the study and its results, the conclusions could be worked up to be more specific

References

Please follow the indications of the Instructions for Authors for formatting the reference list consistently!

Author Response

Many thanks to the reviewers for their time spent on this paper and suggestions for improving it. We have amended the paper based on your suggestions, and appreciate your comments.

We hope that you feel it is improved as a result.

Yours faithfully

Tamzin Furtado (and other authors)

Reviewer 1:

A timely study, taking the advantage of diving in the ongoing situation. Please see bellow some recommendations and comments, with the hope to further help increasing both the value of the paper and clarity for the reader.

Each abbreviation should be spelled out at first usage, even for the consecrated ones such as ‘UK’

amended

Simple summary

Even if the Simple summary is for the general reader, multiple repetition of ‘we’ draws attention and places emphasis on the authors. Please consider using impersonal mode

Amended

L12: please consider rewording “everything was already ‘up in the air’”

Altered to “subject to change”

L15: please insert ‘The’ before “Results’ (specific results of the study)

 Amended

Abstract

L25: ‘necessary’ may be smoother to read

Amended

L27: please consider to change (total 48 interviews) to (48 interviews in total) or (n=48) or (a total of 48 interviews)

Amended

L30: the word ‘ideas’ sounds weak; does it refer to knowledge and experience?

Good point, thank you– changed to “construction of”

L31: please consider to change ‘on’ to ‘in’ after ‘contentment’

Removed – have changed this suggestion when possible but “on yards” is common usage in this context and “in yards” sometimes unfamiliar (as with farms – we would say “50 cows are kept on the farm” and not “in the farm”) so this has not always been upheld. In the specific instance mentioned here I have simply removed it.

L36: ‘instead’ is not clear enough. Please rephrase (LYO/Ms adopted new, movement influencing measures, they limited the horse-management changes to minimal – these are already two slightly/partly contradicting statements. Then the LYO/Ms expectation is stated, for human compliance with the official/legal guidelines, instead of what?)

It’s instead of changing things for the horses: “During this time, LYO/Ms reported prioritising equine wellbeing by limiting change to equine routines and management wherever possible. There was instead, an expectation that the lives of humans would be moulded and re-shaped to fit with the government COVID-19 guidelines.”

Altered to: Instead of altering equine management, there was an expectation that the lives of humans would be moulded and re-shaped to fit with the government COVID-19 guidelines.

L36: ‘the’ before ‘humans’ is not necessary (as long as it refers generally to people, humans)

Amended

L37: please correct ‘hese’ to ‘These’

Amended

L38: please consider to change ‘individual yard’ to ‘each individual yard’ or ‘individual yards’

Amended

L38: please consider to insert ‘good’ between ‘a’ and ‘standard’ (to be clearer).

Altered to maintaining “the” standard care because we are not saying whether the care is good or not – they were maintaining the status quo level of care

Introduction

L43: please consider to change ‘every aspect’ to ‘many aspects’ or ‘most of the aspects’ (from scientifical point of view the ‘every aspect’ statement is neither proven, nor the aim of the article to prove)

Amended

L44: please consider to change ‘prioritise our lives’ to ‘what we do prioritise in our lives’

Rejected due to more complex sentence structure - it would be “…opportunity for reflection about how we structure and what we do to prioritise our lives” rather than “how we structure and prioritise our lives”

L45-46: please consider to change ‘could not be altered’ to ‘unalterable’ or a synonym (alternatively ‘which were the aspects of our daily life we changed and which were those we considered unalterable’ or simpler: ‘our readiness or resistance to change specific aspects of our daily lives’)

Amended

L46: please change ‘consider’ to ‘considered’. Please use simple past tense wherever the performed research (including results) is presented

Rejected stylistic change– we are describing what is happening in this paper, present tense is appropriate

L47: please change ‘on’ to ‘in’ or ‘at’

Amended

L50-51: citation needed for ‘predominantly at livery yards’

Amended, added Hockenhull demographic info paper

L50: same as for L47

Amended

L50: please, here and throughout the paper, follow the requirements of Instructions for Authors regarding the formatting of the reference numbers given in the text and all other requirements stated, as well

Amended

L52: ‘alongside other horse owners’ is redundant and even misleading, please remove

Amended (although they are alongside other horse owners, which is relevant – but discussed later in results so removed here)

L54: same as for L47 here and throughout the paper

Amended

L61: please remove the semicolon

Amended

L64-69: wording can be changed to be shorter and more efficient, especially in L64-67 sentence

Slightly altered to: Many horse owners reported difficulties managing equine obesity because of the rules designated by the LYO/M: for example not being allowed to implement changes in order to effectively manage overweight horses (such as using electric fencing to reduce grazing)6.

L70-72: with respect, in my opinion, giving four references (7-10) to a reasonably general sentence, without citing the respective titles anywhere else in the paper is not justified and may be perceived as incorrect with regards to the reader. Please correct this here and where it applies throughout the paper

This is standard usage and references are given. No change made but happy to discuss if editor prefers change

L72: please be more specific regarding prevalence

Added “perceived” – we are not talking about specific prevalence here so cannot add numbers, but instead the para above has talked about general issues. Cannot therefore add specific prevalence. The issue is about the perceived prevalence, hence change

L73: it would be of a great benefit for the international reader to have a clearer image of the number/percentage of leisure horses kept at livery yards in the UK, please include this information

Added “thought to be over 50%” with Hockenhull (2015) ref

L79: please spell out abbreviations at their first usage

Amended

L80: please be more specific regarding the fraction of approved horse livery yards in England and the overall number

Cannot give overall number as this is unknown; added number of approved yards (950)

L81: please be more specific if ‘each yard’ refers to the approved or non-approved ones

Added “each yard in the UK”

L93: please insert ‘on’ between ‘as’ and ‘any’

amended

 L98: please be consistent in tense usage within the sentence (parallelism)

amended

L99: please consider to change ‘to practice’ to ‘of their practices’; consider to change ‘footfall’ to ‘circulation’ or ‘movement’ with or without specification of humans; on – in – at

 amended

Materials and Methods

L109: please replace ‘COVID-19’ with ‘the COVID-19 pandemic’; here and throughout the paper: please be consistent in the usage of chosen terms

amended

L117: please consider to change ‘that they might be interested in being involved’ to ‘to participate as well’ or similar

amended

L119: same as L109

amended

L124: please correct the beginning of sentence with a number

amended

L132: please consider to use ‘secured’ instead of ‘secure’

amended

L137: please consider to change the word ‘naturalistic’ to a more specific one

Amended to observational

L157: please specify what ‘strongly’ represents

removed

Table 1, please be consistent with the usage of periods (.) and also phrasing. The general aspect of the table is untidy. Inconsistencies of abbreviations which, additionally, are not spelled out in the table’s subscript. The content is for the reader, in my opinion it does not have to respect exactly the formulation (wording, phrasing, citation of results) of the working documents. In certain places rewording can shorten, sharpen, and elevate the meaning/content, please consider to do so. Some (but not all) recommendations:

Apologies - amended

First row: please rephrase ‘if need be.’

amended

Second row: please reword ‘poo’

amended

Third row: contractions should be avoided in academic writing; please rephrase ‘only come rarely anyway’

amended

Sixth row: please correct ‘antibac’

amended

Seventh row: see above

amended

Eighth row: see above

amended

Ninth row: please reword ‘limited people’

amended

Tenth row: ‘No changes other than no riding, sometimes cared for by buddy rather than their owner’ is unclear, please clarify; the same applies for ‘Considered turning them away but liveries didn’t like the idea – had another viable option so took that’; ‘(do your horse and go)’ is redundant; please reword ‘reps’

amended

13th row: ‘it’s been straight forward’ non-academic and redundant

amended

14th row: please reword ‘sep’; reword ‘comp’ amended

17th row: please reword ‘nec.’

amended

20th row: ‘whole’ is redundant; please reword ‘pro’ rider (professional – not necessarily relevant)

Amended “whole”, amended professional rider as this is relevant

22nd row: previous number of horses not relevant; ‘to another lady who does’ irrelevant;

amended

23rd row: please correct capital letter within sentence

amended

24th row: unclear formulation in third column

amended

Results

Understanding fully that the article describes a qualitative research and no statistical analysis was performed/needed, I do consider that numerical results instead of terms such as often, some, most, many, not always, would increase both the value of the paper and the appreciation of the readers, also because those data are available to the authors. Please do consider this opinion for the comments which relate bellow.

Amended where possible

L177: please clarify ‘change in their role from management towards leadership’ meaning from the paper’s point of view: move the definition from the Discussion section to the Introduction or Materials and method so as the reader can have form the beginning clearly defined terms

Understood – we did this in previous versions and readers previously suggested it was out of place in results and better in discussion – have therefore altered to: “led to a change in their role from management of the status quo, towards strategizing, communicating and leading during a time of transformation because of the COVID-19 pandemic.”

Figure 1 caption: capitalize first letter, don’t capitalize P in ‘Pre-COVID-19’

amended

L184-186: the sentence repeats information given in the figure, please correct

The text is given to explain the diagram in a little more detail, before the full results; this is common practice and we feel would be more confusing is removed. Will defer to editorial input here

L192: please be more exact on ‘most yards’

Will leave this one (this is common practice in qual and in this instance we are talking very generally so explaining would need to be extensive – not suitable at this point in results). However, have altered similar terms later

L196-199: repetition of same information

Altered

L202-204 and in the followings: please consider to use quotation marks for citations. Please be consistent (and grammatically correct) with the use of periods. Personally, I do not see the reason for the use of square brackets instead of simple brackets. At first usage of ‘P’ it has to be spelled out

Editorial input on quotation marks? We had followed previous examples in Animals? Happy to add or leave as is. Amended first instance of P (participant)

L210: please consider to change ‘could’ to ‘should’ or ‘must’, ‘have to’

amended

L212: no reason for using italics, please correct

This is subheading – it is consistent with other subheadings at that level

L230: it seems debatable if they ‘had to allow for’ of ‘were subjected to’; maybe consider to use another term

Changed to “allowed for” – this was what the livery yard managers discussed incorporating into their management

L238: please provide numerical data as scientific result, instead of terms such as ‘minority’

Amended throughout wherever appropriate

L241: please do not capitalize ‘pr

amended

L251-252: several repetitions of this statement, please correct

amended

L253-254: redundant sentence (already stated before). Please consider to rephrase the first two paragraphs as the next one (L255-L261) revolves around the same meaning

Amended – removed redundant sentences

L268-273: this is not a result, please move to the Materials and Method section

Apologies – it’s the way it’s written. Have altered it to explain that we are describing the parifcipants’ concerns.

L280: please consider previous comment regarding the use of square brackets and presentation of the participant’s declarations, however, if a parenthesis was ‘opened’ it has to be ‘closed’ too

Amended

L286: not clear for the reader what ‘[sic]’ means here and in the followings as well

Amended

L289: please be consistent in the usage of COVID-19 related terms

Amended

L304-305: please correct punctuation

Amended

L308: here and elsewhere, please use ‘to’ to complete the usage of ‘from’ (e.g. from DIY to full)

Amended

L317-318: please be more specific and use numerical results instead of terms such as ‘often’ and ‘short time periods’

Altered throughout whenever possible or appropriate

L322-324: repetition of previously stated concept, please correct

Amended – removed redundant sentence

L325-327: consider, instead of indicating here what you are going to describe, to change the title of the following subchapter, to be suggestive enough by itself

removed

L328-330: please correct repetition (information stated previously)

removed

L360: please insert a space after the brackets

removed

L362: please correct repetition of ‘burden’ and regarding the three references given consider the comment from L70-72

removed

L364: please consider comment for L317-318 (‘some’)

L368: see above (‘most’)

L380-382: redundant, please remove (summarizing results belongs to the Discussions section, indicating what will follow is not needed)

removed

L417: please be consistent in font usage (italics)

This is subheading – it is consistent with other subheadings at that level

L433: how many participants’?

Amended: “Eleven of the yards which commonly used part-time stabling chose not to take this risk of increasing turnout time, while others felt it was permissible for a short time”

L487: please move the definition of streamlined care if not in the Materials and Method section (best), at least before its first mentioning

Amended to add in methods

L520: please use past tense

amended

L545: please use numerical result instead of ‘most’ and ‘some’

amended

L556: please use past tense

amended

L565: please give numerical result instead of ‘not always’

Can’t give numerical one here as we’re talking about individual horses rather than yards – altered to clarify: “hence was not considered to be an appropriate option for every horse”

L588: please correct the number of period

amended

L589: please correct the number of periods, please be consistent in usage of ellipsis (Is it an ellipsis and a period? Not consistent with previous usages of ellipses)

amended

L593-599: paragraph not belonging to Results section, please correct/move

Disagree – this is summing up results before moving on to discussion. Request to leave as is

Discussion

L611: please be consistent in the usage of citation marks (at four previous usages ‘status quo’ was not between quotation marks), consider widespread advice to write Latin terms in italics / check Instructions for Authors

Amended but have not italicised status quo, given widespread usage (commonly in Animals this is not capitalised; info not included in Instructions for Authors)

L613: please use past tense; please insert ‘According the participants’ answers’ or similar, before ‘each yard’

amended

L619: instead of giving three references for a general and short statement, please explore more the discussion possibilities regarding the respective scientific studies

This refers to the issues previously discussed in introduction; I have added “described in the introduction”

L624: please correct the font in congruency (italics)

Amended

L627-632: please rephrase the citation to avoid the use of italics, ellipses and incongruent periods. Move the definition (see comment for L177)

Have slightly altered, but request to leave the remainder as is for clarity (other reviewers were happy with this), see previous comment re: definition

L644: please correct italics, here and throughout the paper as it is not a conventional way of emphasizing content (assuming that this was the motive for using it)

amended

L655: brackets are not needed (but their content yes)

amended

L661: citation of other studies needed

Added

L676: the small number of LYO/Ms included in the study, compared to their overall numbers in the UK is another limitation

Qualitative research does not aim for generalisability and there has been a call for papers not to cite this as a limitation with this methodology (given qual seeks depth not breadth)

L680: respect the Instructions for Authors for formatting citations

 amended

Conclusions

Based on the study and its results, the conclusions could be worked up to be more specific

Have added further text, particularly around supporting LYO/Ms in areas other than equine welfare

References

Please follow the indications of the Instructions for Authors for formatting the reference list consistently!

Reference formatting altered

Reviewer 2 Report

Tbe use of covid 19 in this study is speculative. The methodology of the study is not appropriate for a scientific paper, the authors did not personally collected the yards data but used puctures send by owner that can have alter the reality. The manuscript is not suitable for publication

Author Response

Dear Reviewer

Many thanks for your time considering this paper. The methodology is well-used in qualitative study, where interviews, pictures and other media form the data used in the study. We have previously published a manuscript in the Equine Veterinary Journal (https://beva.onlinelibrary.wiley.com/doi/abs/10.1111/evj.13436) which hopes to clarify this for reviewers who may be less familiar with such methods. One specific concern was about people potentially altering the images before sending them. Qualitative research methods seek to understand the experiences participants present, which is different to understanding an "absolute truth" in this way: please see the above paper for more information. We have therefore conducted the study in line with recognised and common procedures, but would be happy to discuss your concerns further if need be.

Thank you and kind regards

Tamzin 

Reviewer 3 Report

The objective of this study is very interesting and with utmost important information limited for a specific part of the equine industry.

Is an easy to read text based in interviews using the grounded theory methodology. Has a good design of qualitative research and this is maybe why I find the results section difficult to follow and sometimes with repetitive information which maybe can be better grouped and organized Please don’t understand this as a harsh review; I try to be extremely respectful with scientific activities and research and I have focused on being
constructive towards improving this paper towards publication as there are
certainly data in this paper worthy of publishing.

A few detailed issues:

Line 37: These results

Line 68:  Two spaces between many owners

Line 182: Conceptual

Line 360: ...Service) by

Line 362: Delete burden

Author Response

Many thanks to the reviewers for their time spent on this paper and suggestions for improving it. We have amended the paper based on your suggestions, and appreciate your comments.

We hope that you feel it is improved as a result.

Yours faithfully

Tamzin Furtado (and other authors)

Specific comments:

Thank you for your kind comments. The other reviewer also commented about some sections being repetitive; we have therefore carefully been through the text and removed several areas of this sort, which we hope will clarify and allay your concerns. This has particularly been applied to the first sections in the results, which we realise presented the same information repetitively. This has been altered and we feel the manuscript is improved as a result; thank you for your comments.

A few detailed issues:

Line 37: These results

amended

Line 68:  Two spaces between many owners

amended

Line 182: Conceptual

amended

Line 360: ...Service) by

amended

Line 362: Delete burden

 amended

Round 2

Reviewer 2 Report

I am sorry, but as the previous version I think the manuscript is not suitable for pubblication. The manuscript has not improved